# STBench: Assessing the Ability of Large Language Models in Spatio-Temporal Analysis

## Abstract

The rapid evolution of large language models (LLMs) holds promise for reforming the methodology of spatio-temporal data mining. However, current works for evaluating the spatio-temporal understanding capability of LLMs are somewhat limited and biased. These works either fail to incorporate the latest language models or only focus on assessing the memorized spatio-temporal knowledge. To address this gap, this paper dissects LLMs' capability of spatio-temporal data into four distinct dimensions: knowledge comprehension, spatio-temporal reasoning, accurate computation, and downstream applications. We curate several natural language question-answer tasks for each category and build the benchmark dataset, namely STBench, containing 15 distinct tasks and over 70,000 QA pairs. Moreover, we have assessed the capabilities of 13 LLMs, such as GPT-4o, Gemma and Mistral. Experimental results reveal that existing LLMs show remarkable performance on knowledge comprehension and spatio-temporal reasoning tasks, with potential for further enhancement on other tasks through in-context learning, chain-of-though prompting, and fine-tuning. The code and datasets of STBench are released on
`https://anonymous.4open.science/r/STBench-14C2`.

## 1 Introduction

The rapid advancement of large language models (LLMs) has opened up new possibilities across various domains (Wang et al., 2024; Thirunavukarasu et al., 2023; Zhao et al., 2023). One promising direction is enhancing spatio-temporal data analysis with the ability of LLMs (Li et al., 2024b; 2023; Manvi et al., 2023). Spatio-temporal data, characterized by both spatial and temporal dimensions, encompasses a variety of datasets crucial for many fields such as geography, meteorology, transportation, and epidemiology. Despite LLMs' remarkable proficiency in language-related tasks, their applicability and effectiveness in handling spatio-temporal data remain relatively unexplored.

Existing evaluations of spatio-temporal data fall in two categorizes. The first category (Shi et al., 2022; Mirzaee & Kordjamshidi, 2022; Li et al., 2024a) focus on evaluating the spatial analysis capability of LLMs and design QA pairs of spatial reasoning such as asking "Is the yellow apple to the west of the yellow watermelon?". The QA pairs are constructed in toy environments without temporal information, which is insufficient to assess the ability of LLM on real spatio-temporal tasks. The second category (Gurnee & Tegmark, 2023; Yamada et al., 2023) aims to evaluate the spatio-temporal analysis capability but only assesses the abilities of LLMs' in one specific dimension. For example, the most recent work (Gurnee & Tegmark, 2023) tends to evaluate the memory ability of spatio-temporal knowledge. For a comprehensive evaluation, we argue that the abilities of LLMs in spatio-temporal analysis should contain not only the memory ability but also other dimensions, such as reasoning and knowledge comprehension.

To achieve this goal, we propose a framework, namely STBench, for evaluating the spatio-temporal capabilities of LLMs. As shown in Figure 1, STBench dissects the LLMs' capacity into four distinct dimensions: knowledge comprehension, spatio-temporal reasoning, accurate computation, and downstream applications. **Knowledge Comprehension** examines the model's capacity to understand and interpret the underlying meaning and context of spatio-temporal information. **Spatio-Temporal Reasoning** evaluates the ability to understand and reason about the spatial and temporal relationships between entities and events. **Accurate Computation** handles the precise and complex calculations of

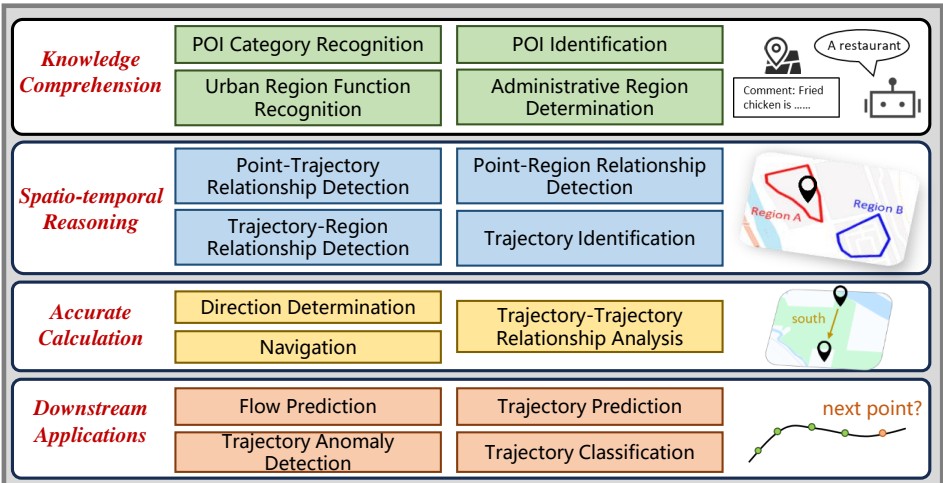

Figure 1: Overview of STBench. It consists of 15 distinct tasks covering four dimensions: knowledge comprehension, spatio-temporal reasoning, accurate calculation and downstream applications.

spatio-temporal data. Moreover, we also employ some **Downstream Applications** such as trajectory anomaly detection and trajectory prediction to assess the ability of LLMs on practical tasks.

For each evaluated dimension, we design several tasks and construct QA pairs to assess the ability of LLMs qualitatively. We have curated a benchmark dataset, STBench, which contains over 70,000 QA pairs and 15 distinct tasks covering the four dimensions. Furthermore, we evaluated the latest 13 LLMs, including GPT-4o[1], Gemma (Mesnard et al., 2024), Llama2 (Touvron et al., 2023), and provide a detailed report that quantitatively assesses the four dimensional abilities of LLMs. Our experimental results reveal that existing LLMs show remarkable performance on knowledge comprehension and spatio-temporal reasoning tasks, but the performance across most models is generally low for accurate computation tasks and downstream application tasks. We also conduct experiments to investigate if in-context learning, chain-of-thought prompting and supervised fine-tuning can enhance the performance of LLMs on spatio-temporal reasoning. The results demonstrate the great potential of LLMs in spatio-temporal data analysis. While numerous benchmarks for knowledge comprehension (Wang et al., 2019), commonsense reasoning (Sakaguchi et al., 2021) and mathematical calculation (Cobbe et al., 2021) have indeed become targets for LLMs to excel and improve upon, the critical area of spatio-temporal data analysis is overlooked. A dedicated benchmark like STBench will not only facilitate the assessment of current models but also encourage further research on spatio-temporal capabilities while developing new LLMs.

The contributions of this paper are summarized as following:

- This paper presents STBench, a comprehensive benchmarking framework designed to evaluate the spatio-temporal analysis capabilities of LLMs. STBench is both user-friendly and highly extensible, allowing users to effortlessly reproduce experimental results across 13 LLMs and 15 tasks with a single script. Its modular design facilitates the seamless addition of new LLMs, tasks, or datasets.

- For a comprehensive evaluation, STBench categorizes spatio-temporal abilities into four dimensions, each with multiple tasks tailored to various data types, including POI, trajectory, region and traffic flow. STBench further incorporates multiple enhancement methods, including in-context learning, chain-of-thought and supervised fine-tuning, to investigate the potential of LLMs in spatio-temporal analysis.

- Extensive experiments are conducted and the results highlight the remarkable performance of LLMs in knowledge comprehension and spatio-temporal reasoning tasks, while also identifying areas for improvement in accurate computation and downstream applications. It reveals the great potential of LLMs in spatio-temporal analysis.

---

[1]https://platform.openai.com/docs/models/gpt-4o

## 2 RELATED WORK

The rapid development of large-scale language models has attracted widespread interest from various communities (Zhang et al., 2024b; Jin et al., 2023; Zhang et al., 2024a; Kasneci et al., 2023). Many researchers studied the capabilities of LLMs (Chang et al., 2023; Chen et al., 2021; 2024) and some of them investigated the potential in spatio-temporal mining.

**Spatial analysis capabilities.** (Mirzaee et al., 2021) proposed a question-answering (QA) benchmark for spatial reasoning with natural language texts. (Shi et al., 2022) presented a QA dataset to evaluate language models' capability of multi-hop spatial reasoning. (Mirzaee & Kordjamshidi, 2022) provided two datasets about spatial question answering and spatial role labeling problems. (Li et al., 2024a) further improved a previous benchmark to provide a more accurate assessment. However, these works only focus on spatial reasoning in toy environments. They ignore the temporal dimension and are far from the real scenarios of spatio-temporal applications.

**Spatio-temporal analysis capabilities.** (Ji & Gao, 2023) evaluated the ability of LLMs to represent geometric shapes and spatial relationships. (MOONEY et al., 2023) examines the performance of ChatGPT in a geographic information systems exam to evaluate its spatial literacy. (Roberts et al., 2023a) investigates the geographic capabilities of GPT-4 (OpenAI, 2023) through a series of qualitative and quantitative experiments. (Gurnee & Tegmark, 2023) analyzes the learned representations of several spatial and temporal datasets by training linear regression probes. (Yamada et al., 2023) evaluates the ability of LLMs to represent and reason about spatial structures, such as squares and hexagons. (Hochmair et al., 2024) assesses four closed-source LLMs on a set of tasks, primarily focusing on coding capabilities, such as code interpretation and code generation. These works either only analyze a specific model or only examine the capabilities of a specific aspect, failing to provide a comprehensive evaluation of the latest closed-source and open-source LLMs. There are two most relevant works and one of which is (Roberts et al., 2023b), which assesses the geographic and geospatial capabilities of multimodal LLMs. Their tasks are completely designed for multimodal models and are not applicable to single-modal large language models. The other one is (Feng et al., 2024), which design 7 tasks in 2 categories of perception-understanding and decision-making to evaluate the capability of LLMs. To comprehensively assess the spatio-temporal ability of LLMs, in this paper, we categorize the spatial-temporal abilities into four dimensions: knowledge comprehension, spatio-temporal reasoning, accurate computation and downstream applications. Based on this, we propose a benchmark consisting of 15 tasks and over 70,000 QA pairs. We benchmarked 13 latest LLMs to assess their capabilities and to investigate their potential in spatio-temporal mining.

## 3 PRELIMINARY

In spatio-temporal data mining, concepts such as Point of Interest (POI) and trajectory play a fundamental role in representing and analyzing spatio-temporal data. Before presenting the construction methodology of our benchmark, we formally define these concepts in this section.

**DEFINITION 1** (Point of Interest): *A point of interest (POI) is a specific geographic location $p = < i_p, lat_p, lon_p, c_p, \mathcal{M}_p >$, where $i_p$ is the ID number, $lat_p$ is the latitude, $lon_p$ is the longitude, $c_p$ denotes the category of this POI and $\mathcal{M}_p = \{m_1, m_2, \cdots\}$ is a set of comments about this POI.*

**DEFINITION 2** (Trajectory): *Each trajectory $t = < t_1, t_2, \cdots >$ is a sequence of points, where each point $t_i = < lat_i, lon_i, time_i >$ is a triplet of latitude, longitude and timestamp.*

**DEFINITION 3** (Region): *A region is a defined area that is distinct from its surroundings. Each region $r = < b_r, c_r, \mathcal{P}_r >$ is characterized by its boundary lines $b_r$ and the region function category $c_r$. The set $\mathcal{P}_r = \{p_1, p_2, \cdots\}$ denotes the POIs that fall in this region.*

**DEFINITION 4** (Inflow/Outflow): *The inflow $I_i^r$ and outflow $O_i^r$ are defined as the number of trajectories entering and leaving a specific region $r$ within the $i$-th time interval, respectively.*

Table 1: A prompt template of the samples in STBench. The blue texts describe the question. The brown texts are the options. The teal texts denote the guidance that constrains the output of LLMs.

---

Question: Below is the coordinate information and related comments of a point of interest: $\cdots$.
Please answer the category of this point of interest.
Options: (1) xxxx, (2) xxxx, (3) xxxx, $\cdots$.
Please answer one option.
Answer: The answer is option (

---

## 4 BENCHMARK CONSTRUCTION

In this section, we propose a benchmark, STBench, to assess the ability of LLMs in spatio-temporal analysis. We will begin by presenting the considerations that guide the design of STBench. Subsequently, we will delve into a detailed exposition of the construction of STBench.

### 4.1 OVERVIEW

To construct a benchmark for assessing the ability of LLMs in spatio-temporal data, we should first consider the evaluation tasks and the data format.

**Ability Categories**. Choosing or designing appropriate tasks is crucial for assessing the ability of LLMs in spatio-temporal data mining. Real-world applications often require a mixture of multiple abilities, *e.g.*, the POI recommendation task requires both the knowledge comprehension ability to understand the semantics of different POI categories and the spatio-temporal reasoning ability to infer the mobility patterns. Thus it is difficult to separately evaluate each capability dimension of LLMs and analyze their strengths and weaknesses solely based on real-world tasks. Therefore, to provide a comprehensive evaluation, we categorize the requisite abilities into four dimensions: *knowledge comprehension*, *spatio-temporal reasoning*, *accurate computation*, and *downstream applications*. For each category, we design several tasks for assessment.

**Data Format**. Another important question is what data format we should adopt. There are some problems if we directly ask the model through dialogue and allow open-ended answers. Firstly, the response of LLMs is uncontrolled. For instance, models may only apologize for not being able to provide an accurate answer, rather than directly responding to our question. Moreover, open-ended answers make it difficult to identify the final answer of LLMs, *e.g.*, LLMs may reply with a lot of explanation or even some unrelated content. Therefore, we have LLMs complete the input texts, rather than asking LLMs through dialogue. As shown in Table 1, each data sample in STBench consists of three parts: the question, the options and the guidance. The LLMs should continue the guidance text, *i.e.*, they should generate an option number, thus the output is controllable. Note that some chat models do not support text completion, thus we instruct these models to complete the texts through system prompts. The details are in Appendix A in the supplementary material.

### 4.2 KNOWLEDGE COMPREHENSION

The model's capacity to understand and interpret the underlying meaning and context of spatio-temporal information is important. This involves the ability to comprehend the semantic nuances within the data and the knowledge of relevant spatio-temporal concepts and entities, *e.g.*, understanding and distinguishing different POI categories. We provide valuable insights into LLMs' spatio-temporal knowledge comprehension capabilities through four tasks: POI category recognition, POI identification, urban region function recognition, and administrative region determination.

**POI Category Recognition (PCR)**. The semantics of POI are crucial in various applications such as POI recommendation, thus we design this task to evaluate LLM's understanding of POI semantics. Data samples of this task are generated based on the public Yelp dataset[2]. Specifically, we randomly sample some POIs from the Yelp dataset for data construction. For each POI $p = <i_p, lat_p, lon_p, c_p, \mathcal{M}_p>$, we randomly select two comments $m_{i_1}, m_{i_2}$ from the comment

---

[2]https://www.yelp.com/dataset.

set $\mathcal{M}_p$. Then, LLMs are asked to predict the category $c_p$ of the POI according to its coordinates $< lat_p, lon_p >$ and the selected comments $< m_{i_1}, m_{i_2} >$. The POI category $c_p$ and four other randomly sampled POI categories are provided as options.

**POI Identification (PI)**. In this task, the coordinates and comments of two POIs are provided and LLMs are asked to determine if they are the same POI or not. For a POI $\boldsymbol{p} =< i_p, lat_p, lon_p, c_p, \mathcal{M}_p >$ in the Yelp dataset, we construct a positive sample (*i.e.*, the answer is "Yes") and a negative sample based on it. For the positive sample, we ask the model if $< lat_p, lon_p, m_{i_1}, m_{i_2} >$ and $< lat_p + \epsilon_1, lon_p + \epsilon_2, m_{i_3}, m_{i_4} >$ describe the same POI, where $m_{i_j}, 1 \leq j \leq 4$ are comments sampled from the comment set $\mathcal{M}_p$ and $\epsilon_1, \epsilon_2 \sim U(0.0004, 0.0008)$ are minor disturbances to the coordinates. For negative samples, we construct a KD-Tree and sample another POI $\boldsymbol{p}' =< i_{p'}, lat_{p'}, lon_{p'}, c_{p'}, \mathcal{M}_{p'} >$ from the nearest five neighbors of $\boldsymbol{p}$. Then, the negative sample is constructed based on $< lat_p, lon_p, m_{i_1}, m_{i_2} >$ and $< lat_{p'}, lon_{p'}, m_{i_5}, m_{i_6} >$, where $m_{i_5}, m_{i_6}$ are comments sampled from the comment set $\mathcal{M}_{p'}$.

**Urban Region Function Recognition (URFR)**. This task requires LLMs to predict the urban region function according to the boundary lines and the POIs located in the region, which evaluates LLMs' understanding of urban regions. To construct data samples, we first match POIs in the Yelp dataset and regions in the New Orleans region dataset[3], removing POIs that do not fall in any region and regions that contain no more than one POI. After that, for each region $\boldsymbol{r} =< b_r, c_r, \mathcal{P}_r >$, we randomly select two POIs $\{\boldsymbol{p}_k =< i_{p_k}, lat_{p_k}, lon_{p_k}, c_{p_k}, \mathcal{M}_{p_k} > | k = i_1, i_2\}$ from its POI set $\mathcal{P}_r$. For each $p_k$, two comments $m_1^{p_k}, m_2^{p_k}$ are sampled from the comment set $\boldsymbol{M}_{p_k}$, where $k \in i_1, i_2$. Then, we ask LLMs to predict the region function $c_r$ according to its boundary lines $b_r$, the coordinates and comments of the selected POIs, *i.e.*, $\{< lat_{p_k}, lon_{p_k}, m_1^{p_k}, m_2^{p_k} > | k = i_1, i_2\}$. We provide the region function $c_r$ and four other region function categories as options.

**Administrative Region Determination (ARD)**. This task refers to determining which administrative region a coordinate is located in, which involves relevant knowledge of the administrative regions and the ability to associate it with geographical coordinates. For a POI $\boldsymbol{p} =< i_p, lat_p, lon_p, c_p, \mathcal{M}_p >$ of the Yelp dataset located in $city_p$, LLMs are asked to answer which city $< lat_p, lon_p >$ is located in. $city_p$ along with other four cities in the same state are provided as options.

## 4.3 SPATIO-TEMPORAL REASONING

Spatio-temporal reasoning encompasses the ability to understand and reason about the spatial and temporal relationships between entities and events. For example, given a POI and some regions, LLMs should determine which region the POI falls in according to their coordinates and boundary lines. We design four tasks to assess the spatio-temporal reasoning ability of large language models: point-trajectory relationship detection, point-region relationship detection, trajectory-region relationship detection and trajectory identification.

**Point-Trajectory Relationship Detection (PTRD)**. The task is to determine whether a trajectory passes through a point. To generate a data sample, we downsample the trajectory in the public Xi'an dataset[4] into a shorter trajectory $\boldsymbol{t} = \{t_1, \cdots, t_n\}$ and construct five points as options. We take $< (lat_i + lat_{i+1})/2, (lon_i + lon_{i+1})/2 >$ as the true option, where $< lat_i, lon_i >$ and $< lat_{i+1}, lon_{i+1} >$ are two adjacent points in the trajectory. To construct an error option, we sample a point $t_j =< lat_j, lon_j, time_j >$ from the trajectory and perturb its coordinates with Gaussian noise, *i.e.*, the error option is $< lat_j + \epsilon_1, lon_j + \epsilon_2 >$, where $\epsilon_1, \epsilon_2 \sim \mathcal{N}(0.01, 0.001)$.

**Point-Region Relationship Detection (PRRD)**. Given a point and several regions, this task aims to infer which region the point falls in. To generate a data sample, we select $i$ regions $\{\boldsymbol{r}_1, \cdots, \boldsymbol{r}_i\}$ located in the same city from the EULUC dataset Gong et al. (2020). Then, a region $\boldsymbol{r}_j$ is chosen from these $i$ regions and we randomly selected a point $\boldsymbol{p}$ in region $\boldsymbol{r}_j$. The coordinates of point $\boldsymbol{p}$ and the boundary lines of $i$ regions are used to generate the question texts, and all $i$ regions are provided as options. We construct four sub-datasets by varying the value of $i$ from 2 to 5.

**Trajectory-Region Relationship Detection (TRRD)**. Given a trajectory and some regions, this task aims to determine which regions the trajectory has passed through chronologically. To construct a data sample, we randomly select five regions $\{\boldsymbol{r}_1, \cdots, \boldsymbol{r}_5\}$ located in the same city from the EULUC

---

[3]https://catalog.data.gov/dataset/zoning-district-9939c

[4]https://gaia.didichuxing.com/

dataset and generate a trajectory $t$ by a random walk. The region sequence that $t$ passes through and four randomly generated region sequences are provided as options. We construct five sub-datasets by setting the length of $t$ to 2, 4, 6, 8 and 10, respectively.

**Trajectory Identification (TI)**. In this task, we ask LLMs to determine if two point sequences $t'$ and $t''$ are sampled from the same trajectory. We propose two strategies to construct positive samples (*i.e.*, samples with the answer "Yes") and two strategies to construct negative samples. Specifically, for a trajectory $t = < t_1, t_2, \cdots >$ in the Xi'an dataset, we construct two positive samples through downsampling and staggered sampling. For instance, the downsampling strategy use $t' = < t_1, t_2, t_3, \cdots >$ and $t'' = < t_1, t_3, t_5, \cdots >$ to generate the question, while the staggered sampling strategy use $t' = < t_1, t_3, t_5, \cdots >$ and $t'' = < t_2, t_4, t_6, \cdots >$ to generate the question. To construct negative samples, we downsample a trajectory $t$ into $t'$ and add temporal offsets or spatial offsets to $t'$ to obtain $t''$.

## 4.4 Accurate computation

In the context of handling spatial-temporal data, accurate computation plays a pivotal role. It focuses on the model's capability to perform precise and complex calculations related to spatial-temporal data. We include three tasks that challenge the model's accuracy in spatial-temporal computations for assessment: direction determination, navigation and trajectory-trajectory relationship detection.

**Direction Determination (DD)**. This task is to determine the direction between two geographical points. To create a data sample, two POIs are randomly chosen from the Yelp dataset, and the model is asked to calculate the corresponding azimuth and to determinate their relative direction based on the calculation result. Eight options are provided for all data samples: north, south, west, east, northeast, northwest, southeast and southwest.

**Navigation (NAV)**. In this task, LLMs are asked to plan a shortest route from a source point to a destination point based on a given road network. To construct a data sample, we randomly selected $n$ points $p_1, \cdots, p_n$ from a given region, interconnect them to form a complete graph, and subsequently apply Kruskal's algorithm (Kruskal, 1956) to derive the minimum spanning tree $G$ from this complete graph. We add edges to $G$ so that we can obtain a connected graph $G'$ with $1.5n$ edges. We randomly sample two points $p_s$ and $p_d$ as the source point and destination point, and ask LLMs which edge connecting to $p_s$ is on the shortest path. All edges connecting to $p_s$ are provided as options. There are two sub-datasets, i.e., edges with weights and edges without weights, where LLMs need to minimize the length or hop count of the route, respectively.

**Trajectory-Trajectory Relationship Analysis (TTRA)**. This task is to calculate the number of times two trajectories encounter each other. To construct a data sample, we generate two trajectories $t = < t_1, \cdots, t_n >$ and $t' = < t'_1, \cdots, t'_n >$ through random walks within a certain area. We count it as an encounter if $t_i t_{i+1}$ and $t'_j t'_{j+1}$ intersect in space and overlap in time, where $1 \leq i, j \leq n - 1$. We provided the ground truth and other four wrong answers as options.

## 4.5 Downstream Applications

Downstream tasks require the model to not only understand the spatial-temporal context but also apply this understanding to practical applications. We assess this aspect of LLMs through four downstream tasks: flow prediction, trajectory anomaly detection, trajectory classification and trajectory prediction.

**Flow Prediction (FP)**. This task requests LLMs to predict the future inflows and outflows based on the historical inflows and outflows. Specifically, to construct a data sample, we randomly select a region $r$ and a timestamp $t$ from TaxiBJ (Zhang et al., 2017) and ask LLMs to predict $I_{t+i}^r$ and $O_{t+i}^r$ ($1 \leq i \leq 6$) of the next 6 time intervals according to the historical inflows $I_{t-j}^r$ and outflows $O_{t-j}^r$ ($0 \leq j \leq 12$) over the past 12 time intervals.

**Trajectory Anomaly Detection (TAD)**. In order to detect anomalous trajectories, LLMs should infer the underlying route and shape from trajectory data. We consider trajectories in Xi'an dataset as normal and perform detours to generate anomalous samples. Specifically, given a trajectory $t = < t_1, \cdots, t_n >$, we identify the direction perpendicular to the line connecting $t_1$ and $t_n$, and move the middle one-third of the trajectory along this direction to generate an anomalous sample.

**Trajectory Classification (TC)**. This task requires the model to comprehensively consider the coordinates, length, speed and other relevant information to distinguish different trajectories. We construct dataset for this task based on the Geolife dataset[5]. Due to the input length limitation of LLMs, we downsample each trajectory and ask LLMs to infer what generates the trajectory. Three options are provided: bike, car and pedestrian.

**Trajectory Prediction (TP)**. This task is to predict the next point based on the historical points of a trajectory, which involves the ability to model the trajectory patterns and the moving speed. We construct data samples for this task based on the trajectories in the Xi'an dataset. Specifically, we first downsample the each trajectory with a time interval of 30 seconds. Then, for each trajectory $t = < t_1, t_2, \cdots, t_n >$, we ask LLMs to predict the coordinates of $t_j$ according to the historical points $< t_1, \cdots, t_{j-1} >$, where $3 \leq j \leq n$. Note that we do not provide options in this task.

## 5 EXPERIMENTS

We conduct extensive experiments on STBench to evaluate the spatial-temporal ability of LLMs and to investigate if in-context learning, chain-of-thought and fine-tuning can improve the performance.

### 5.1 EXPERIMENTAL SETUP

**Evaluated models**. We evaluate the performance of two closed-source model, *i.e.*, **ChatGPT** and **GPT-4o**, and a set of open-source models: **Llama-2** (Touvron et al., 2023), **Vicuna**[6], **Gemma** (Mesnard et al., 2024), **Phi-2**, **ChatGLM2**, **ChatGLM3**, (Du et al., 2022; Zeng et al., 2023), **Mistral** (Jiang et al., 2023), **Falcon** (Almazrouei et al., 2023), **Deepseek** (Bi et al., 2024), **Qwen** (Bai et al., 2023) and **Yi** (Young et al., 2024). More introduction to these models can be found in Appendix B.1 in the supplementary material.

**Metrics**. We adopt accuracy for tasks other than trajectory prediction and flow prediction. For trajectory prediction, we report absolute error, *i.e.*, the distance in meters between the predicted coordinates and ground truth. For flow prediction, we adopt MAE and RMSE as the metrics.

**Experimental details**. In our experiments, we adopt the precision of FP32 for all LLMs. For all tasks except trajectory prediction, LLMs are expected to answer an option or "Yes"/"No", thus we set the $max\_new\_tokens$ to 15, *i.e.*, the maximum length of the generated new tokens is 15. For trajectory prediction and flow prediction, we set the $max\_new\_tokens$ to 50. For other hyperparameters, we adopt the default value of each model. All experiments of open source models are conducted on two NVIDIA H100.

### 5.2 MAIN RESULTS

To investigate the spatio-temporal ability of LLMs, we conduct experiments to evaluate the performance of all models on each task. The main results are shown in Table 2 and Table 3. More detailed results and analysis, *e.g.*, results regarding each sub-dataset, can be found in Appendix B.2 in the supplementary material.

**Model size is important for knowledge comprehension**. For knowledge comprehension, GPT-4o performs better than ChatGPT on all tasks, and ChatGPT outperforms other models on most tasks. Take PCR as an example, GPT-4o achieved an accuracy of 95.88% and ChatGPT achieved an accuracy of 79.26%, while the accuracy of other open-source LLMs is below 50%. The possible reason is that LLMs rely on sufficient parameters to compress and store knowledge, and ChatGPT/GPT-4o has more parameters than other evaluated open-source models. We also observe that Gemma-2B performs poorly on all knowledge comprehension tasks, while Gemma-7B, with the same technology but more parameters, achieves higher performance. It also supports the conclusion that model size is important for knowledge comprehension.

**The evaluated models have difficulty in multi-step reasoning.** The performance of most models on point-region relationship detection is much higher than trajectory-region detection. For instance,

---

[5]https://www.microsoft.com/en-us/research/publication/geolife-gps-trajectory-dataset-user-guide/
[6]https://lmsys.org/blog/2023-03-30-vicuna/

Table 2: The performance of *ACC* on knowledge comprehension and spatio-temporal reasoning tasks (bold: best closed-source LLM; underline: best open-source LLM). '-' denotes the model failed to answer most questions.

| | Knowledge Comprehension | | | | Spatio-temporal Reasoning | | | |
|---|---|---|---|---|---|---|---|---|
| | PCR | PI | URFR | ARD | PTRD | PRRD | TRRD | TI |
| ChatGPT | 0.7926 | 0.5864 | 0.3978 | 0.8358 | **0.7525** | **0.9240** | 0.0258 | 0.3342 |
| GPT-4o | **0.9588** | **0.7268** | **0.6026** | **0.9656** | - | 0.9188 | **0.1102** | **0.4416** |
| ChatGLM2 | 0.2938 | 0.5004 | 0.2661 | 0.2176 | 0.2036 | 0.5216 | 0.2790 | 0.5000 |
| ChatGLM3 | 0.4342 | 0.5272 | 0.2704 | 0.2872 | 0.3058 | 0.8244 | 0.1978 | 0.6842 |
| Phi-2 | - | 0.5267 | - | 0.2988 | - | - | - | 0.5000 |
| Llama-2-7B | 0.2146 | 0.4790 | 0.2105 | 0.2198 | 0.2802 | 0.6606 | 0.2034 | 0.5486 |
| Vicuna-7B | 0.3858 | 0.5836 | 0.2063 | 0.2212 | 0.3470 | 0.7080 | 0.1968 | 0.5000 |
| Gemma-2B | 0.2116 | 0.5000 | 0.1989 | 0.1938 | 0.4688 | 0.5744 | 0.2014 | 0.5000 |
| Gemma-7B | 0.4462 | 0.5000 | 0.2258 | 0.2652 | 0.3782 | 0.9044 | 0.1992 | 0.5000 |
| DeepSeek-7B | 0.2160 | 0.4708 | 0.2071 | 0.1938 | 0.2142 | 0.6424 | 0.1173 | 0.4964 |
| Falcon-7B | 0.1888 | 0.5112 | 0.1929 | 0.1928 | 0.1918 | 0.4222 | 0.2061 | 0.7072 |
| Mistral-7B | 0.3526 | 0.4918 | 0.2168 | 0.3014 | 0.4476 | 0.7098 | 0.0702 | 0.4376 |
| Qwen-7B | 0.2504 | 0.6795 | 0.2569 | 0.2282 | 0.2272 | 0.5762 | 0.1661 | 0.4787 |
| Yi-6B | 0.3576 | 0.5052 | 0.2149 | 0.1880 | 0.5536 | 0.8264 | 0.1979 | 0.5722 |

the accuracy of ChatGPT is 92.40% on point-region relationship detection, with only 2.58% on trajectory-region relationship detection. Note that trajectory-region relationship detection can be achieved by performing point-region relationship detection for each point in the trajectory, thus it is a multi-step reasoning task. Although models such as ChatGPT, GPT-4o, and Gemma-7B can achieve high performance on each step, their performance on this multi-step task is poor.

**Accurate computation and downstream tasks are more challenging**. As shown in Table 3, the accuracy of all models except GPT-4o is below 45% on accurate computation tasks, which is because LLMs are mainly trained on nature language corpus and are not good at computation. We also find that GPT-4o outperforms other LLMs by a large margin, *e.g.*, it achieved an accuracy of 75.52% on NAV, with a relative improvement of 72.3% compared to other LLMs. This is consistent with the significant improvement in mathematical ability of GPT-4o. Moreover, the performance of evaluated models is also poor on downstream tasks. For instance, the best performance on trajectory anomaly is only 60.16%, indicating that most evaluated models can not distinguish between normal and anomalous trajectories. The lack of expert knowledge on downstream tasks, *e.g.*, the normal trajectory patterns, leads to their unsatisfactory performance.

**A suitable model is more important than larger parameters for spatio-temporal mining.** We observe ChatGPT and GPT-4o outperform poorer than most open-source models on TRRD and TI, despite having a larger number of parameters. On FP, the lightweight model, Phi-2, with only 2.7B parameteres, performs better than all models except Gemma-7B. Although LLMs have the potential to analyze spatio-temporal data, not all models have been adequately trained on relevant corpora and learned corresponding spatio-temporal ability, regardless of the model size. It leads to a significant difference in performance between different models for many spatio-temporal tasks.

## 5.3 IN-CONTEXT LEARNING EVALUATION

Although some evaluated LLMs can perform well on certain tasks, the results in many scenarios are poor. Since LLMs show impressive in-context few-shot learning capacity in previous works, we conduct experiments to investigate if in-context learning can improve the performance of LLMs on STBench. Specifically, we select six tasks where the evaluated models performed poorly and we adopt two-shot prompting. Due to the heavier computation cost caused by the longer context, we only evaluate one closed-source model, ChatGPT, and two open-source models with different model sizes, *i.e.*, Gemma-2B and Llama-2-7B. The results are shown in Fig. 2(a).

Table 3: The performance of *ACC*, *MAE* and absolute error (in meters) on accurate computation and downstream tasks (bold: best closed-source LLM; underline: best open-source LLM). '-' denotes the model failed to directly answer most questions.

| | Accurate Computation | | | Downstream Applications | | | |
|---|---|---|---|---|---|---|---|
| | DD | NAV | TTRA | FP | TAD | TC | TP |
| ChatGPT | 0.1698 | 0.4384 | 0.1048 | **37.33** | 0.5382 | **0.4475** | - |
| GPT-4o | **0.5434** | **0.7552** | **0.3404** | 43.25 | **0.6016** | - | - |
| ChatGLM2 | 0.1182 | 0.2924 | 0.1992 | 63.72 | 0.5000 | 0.3333 | 231.2 |
| ChatGLM3 | 0.1156 | 0.2576 | 0.1828 | 59.24 | 0.5000 | 0.3111 | 224.5 |
| Phi-2 | 0.1182 | 0.2912 | 0.0658 | 34.82 | 0.5000 | 0.3333 | 206.9 |
| Llama-2-7B | 0.1256 | 0.2774 | 0.2062 | 53.79 | 0.5098 | 0.3333 | 189.3 |
| Vicuna-7B | 0.1106 | 0.2588 | 0.1728 | 48.19 | 0.5000 | 0.2558 | 188.1 |
| Gemma-2B | 0.1972 | 0.2592 | 0.2038 | 41.79 | 0.5000 | 0.3333 | 207.7 |
| Gemma-7B | 0.1182 | 0.3886 | 0.1426 | 31.85 | 0.5000 | 0.3333 | 139.4 |
| DeepSeek-7B | 0.1972 | 0.3058 | 0.1646 | 56.89 | 0.5000 | 0.3333 | 220.8 |
| Falcon-7B | 0.1365 | 0.2610 | 0.2124 | 62.52 | 0.5000 | 0.3309 | 3572.8 |
| Mistral-7B | 0.1182 | 0.3006 | 0.1094 | 42.59 | 0.5000 | 0.3333 | 156.8 |
| Qwen-7B | 0.1324 | 0.3106 | 0.2424 | 53.49 | 0.5049 | 0.3477 | 205.2 |
| Yi-6B | 0.1284 | 0.3336 | 0.2214 | 52.03 | 0.5000 | 0.3333 | 156.2 |

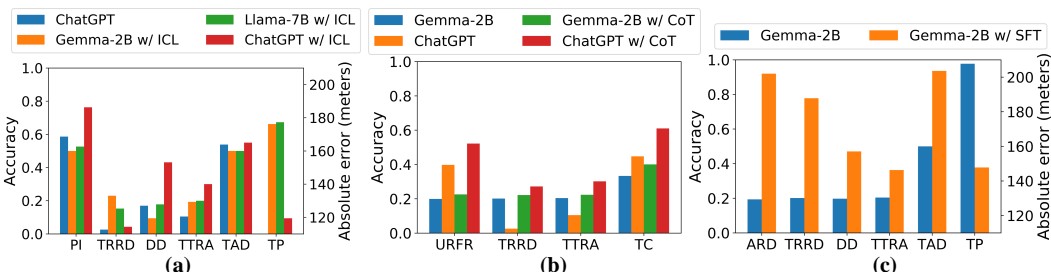

Figure 2: The performance of ACC and absolute error (in meters) in (a) in-context learning evaluation, (b) chain-of-thought evaluation, (c) fine-tuning evaluation.

The performance of ChatGPT has been greatly improved with in-context learning. For instance, its accuracy on POI identification and direction determination has increased from 58.64% to 76.30%, and from 16.98% to 43.16%, respectively. Moreover, the two-shot prompting also constrains the output, *e.g.*, ChatGPT refuses to answer the questions of trajectory prediction in Table 3, but its absolute error is only 119.4 with two-shot prompting. Although in-context learning is effective for ChatGPT, it is useless for Gemma-2B and Llama-2-7B, which is consistent with the phenomenon in previous work that in-context learning is less effective for smaller LLMs (Wei et al., 2022).

### 5.4 CHAIN-OF-THOUGHT EVALUATION

We further conduct experiments to verify if chain-of-thought (CoT) is effective on STBench. Specifically, we evaluate ChatGPT and Gemma-2B with CoT prompting on several tasks that involve multi-step reasoning: urban region function recognition, trajectory-region relationship detection, trajectory-trajectory relationship analysis and trajectory classification. For each task, we add two samples with a detailed reasoning process in the context, *i.e.*, we implement CoT by two-shot prompting. For instance, in trajectory classification, we add two samples that contain the reasoning process of calculating the length and average speed of the trajectory. The results are shown in Fig. 2(b).

We observe the performance of ChatGPT increases significantly in all selected tasks. For instance, its accuracy with CoT prompting is 52.20% on URFR and 61.04% on TC, much better than 39.78% and 44.75% in Table 2 and Table 3. For Gemma-2B, the performance on all selected tasks is also improved. For example, its accuracy increased from 19.89% to 22.55% on urban region function

|  |  | Phi-2 | Gemma-2B | Gemma-2B w/ SFT | STID | PatchTST |
|---|---|---|---|---|---|---|
| Inflow | MAE | 38.14 | 41.39 | 26.79 | 38.57 | 24.43 |
|  | RMSE | 42.59 | 45.54 | 30.87 | 43.62 | 28.28 |
| Outflow | MAE | 31.50 | 42.19 | 25.91 | 36.96 | 23.49 |
|  | RMSE | 35.80 | 46.21 | 29.87 | 42.04 | 27.25 |

Table 4: The performance of *MAE* and *RMSE* of Phi-2, Gemma-2B, fine-tuned Gemma-2B, STID and PatchTST.

recognition and from 33.33% to 40.05% on trajectory classification. The results demonstrate the effectiveness of CoT prompting in spatio-temporal analysis.

### 5.5 FINE-TUNING EVALUATION

While in-context learning and chain-of-thought is less effective for smaller models, we conduct experiments to investigate if fine-tuning can significantly improve the performance on STBench. Specifically, we select several tasks and follow the construction strategies in Section 4 to generate 1,2000 samples as the training dataset for each task. We adopt QLoRA Dettmers et al. (2023) to fine-tune the model on the training dataset for each task, with the learning rate of 2e-4, the rank of 8 and NF4 quantization. Due to the very high computational cost and memory usage, we only fine-tune a 2B model for evaluation, *i.e.*, Gemma-2B. To compare the fine-tuned LLM with existing supervised methods, we train two effective flow prediction method, *i.e.*, STID (Shao et al., 2022) and PatchTST (Nie et al., 2023), on the same dataset. The results are shown in Fig. 2(c) and Table 4.

The performance on all tasks in Fig. 2 is significantly improved after fine-tuning. For instance, the accuracy on administrative region determination and direction determination increased from 19.89% to 91.98%, and from 19.72% to 47.08%, respectively. For trajectory prediction, Gemma-2B achieves the absolute error of 147.8 meters, which is better than all 7B models in Table 3. This confirms LLMs' potential in spatial-temporal analysis and existing LLMs' lack of training on relevant corpora.

As shown in Table 4, the zero-shot capability of LLMs is surprising that Phi-2 (without fine-tuning and few-shot prompting) can surpass the supervised method STID. While Gemma-2B performs poorer than both STID and PatchTST, it outperforms STID and achieved comparable performance to PatchTST after supervised fine-tuning. Overall, the experimental results reveal the bright prospects of LLMs in spatio-temporal data analysis.

## 6 CONCLUSION

In this work, we propose STBench to assess LLMs' ability in spatio-temporal analysis. STBench consists of 15 tasks and over 70,000 QA pairs, systematically evaluating four dimensions: knowledge comprehension, spatio-temporal reasoning, accurate computation, and downstream applications. We benchmark 13 latest LLMs and the results show their remarkable performance on knowledge comprehension and spatio-temporal reasoning tasks. Our further experiments with in-context learning, chain-of-thought prompting and fine-tuning also prove the great potential of LLMs on other tasks.

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
