APPENDIX

## A  DATA FORMAT

### A.1  PROMPT TEMPLATE FOR CHATTING MODELS

To make the responses of LLMs controllable and identification of the final answer easier, all data samples in STBench are constructed in the form of text completion. However, there are some chatting models that only support chat completion and do not support text completion, *e.g.*, GPT-4o. For these models, we instruct them to complete the text entered by the human via system prompt. The data samples we constructed are inputted with the role of human, as shown in Table 5.

Table 5: A prompt template of models that only support chat completion. The blue texts describe the question. The brown texts are the options. The teal texts denote the guidance that constrains the output of LLMs.

| |
|---|
| **System**: "You are a helpful text completion assistant. Please continue writing the text entered by the human." |
| **Human**: "Question: There is a trajectory, xxxx. Options: (1) xxxx, (2) xxxx, (3) xxxx, ⋯. Please answer one option. Answer: The answer is option (" |

### A.2  DATA EXAMPLES

STBench consists of 15 distinct tasks, covering four dimensions: knowledge comprehension, spatio-temporal reasoning, accurate computation and downstream applications. We will provide data samples to illustrate the design of each task.

### A.2.1  KNOWLEDGE COMPREHENSION

There are four tasks to assess the knowledge comprehension ability of LLMs in spatio-temporal analysis, *i.e.*, **Administrative Region Determination (ARD)**, **POI Category Recognition (PCR)**, **POI Identification (PI)** and **Urban Region Function Recognition (URFR)**.

As shown in Table 6, for administrative region determination, we provide the coordinates of a location and ask the model to answer which option the coordinates is located in. The options contain five cities in the same state, which makes this task more challenging. The data sample of POI category recognition is shown in Table 7. LLMs are asked to predict the category of the POI according to its coordinates and two comments, where each comment contains the comment content and the timestamp. We provide five options and each option is a list of tags such as shopping and skin care. In POI identification, we ask the model if two POI are actually the same, where the description of each POI consists of its coordinates and two comments, just as shown in Table 8. A data sample of urban region function recognition is presented in Table 9, which asks the model to predict the urban region function category according to its boundary lines and the POIs located within it.

Table 6: A data sample for ARD. The blue texts describe the question. The brown texts are the options. The teal texts denote the guidance that constrains the output of LLMs.

| | |
|---|---|
| Question | Question: Below is the coordinate location information, and the options of the area where the coordinate may be located:
{
    "latitude": 36.104588,
    "longitude": -86.81415,
    "options": "(0): Eaton, TN (1): Nashville, TN (2): Sewanee, TN (3): Memphis, TN (4): Knoxville, TN"
}
Please answer which area the coordinate is located in. Please just answer the number of your option with no other texts.
Answer: Option ( |
| Answer | 1): Nashville, TN |

Table 7: A data sample for PCR. The blue texts describe the question. The brown texts are the options. The teal texts denote the guidance that constrains the output of LLMs.

| | |
|---|---|
| Question | Question: Below is the coordinate location information and related comments of a location, with the options of possible function for this location:
{
    "latitude": 36.0423589,
    "longitude": -86.7788876,
    "comment1": {
        "content": "BEST WAX CENTER. ZOIE AND ERIN ARE THE BEST. zoie is calm and friendly makes me feel comfortable all the time erin kills the game with my eyebrows every single time. Everyone asks about my eyebrows thanks to her. Definitely recommend going to Erin and zoie.",
        "time": "2021-06-02 00:37:48"
    },
    "comment2":{
        "content": "I have done 2 sessions with Erin and LOVE HER! I was so incredibly nervous my first time getting a full bikini wax, but she made me feel so comfortable. She talked through what she was doing, asked me questions, and made the process seem less painful overall. I highly recommend her!!",
        "time": "2021-06-18 00:28:02"
    },
    "options": "(0): Computers, Shopping, Appliances, Furniture Stores, Home & Garden (1): Waxing, Hair Removal, Skin Care, Beauty & Spas (2): Juice Bars & Smoothies, Food, Vegan, Restaurants, Acai Bowls (3): Discount Store, Shopping, Toy Stores, Food, Candy Stores, Specialty Food (4): Delis, Food, Coffee & Tea, Sandwiches, Restaurants, Convenience Stores"
}
Please answer which function the location is. Please just answer the number of your option with no other texts.
Answer: Option ( |
| Answer | 1): Waxing, Hair Removal, Skin Care, Beauty & Spas |

Table 8: A data sample for PI. The blue texts describe the question. The brown texts are the options. The teal texts denote the guidance that constrains the output of LLMs.

| | |
|---|---|
| Question | Question: Below are two Points of Interest (POI) and related comments.
POI 1:
{
    "latitude": 34.4266787,
    "longitude": -119.7111968,
    "comment1": {
        "content": "Abby Rappoport helped me achieve a long lost sense of health. I was suffering from debilitating insomnia due to a very stressful job and family requirements. She also was able to get me through a bad bout of bronchitis. She is professional, thorough and clearly seasoned as a healthcare provider. I highly recommend Abby if your situation needs caring attention.",
        "time": "2012-08-09 20:43:27"
    },
    "comment2": {
        "content": "Abby is an amazing practitioner. In a treatment she is really present with me and my concerns. She is caring and thorough. I especially appreciate the exercise, herbs and advice she sends me home with so that my healing can continue outside her office. Abby has helped me with stress related problems and chronic low back pain. Sadly, she moved out of my area but whenever I'm her neck of the woods I take the opportunity to see her.",
        "time": "2013-03-01 06:11:05"
    }
}.
POI2:
{
    "latitude": 34.4266621,
    "longitude": -119.711207,
    "comment1": {
        "content": "Before buying I looked to see if they had a map off merchants to see where they were located and found no map. If there is one there out is hard to find. I won't buy unless I can tell if members are near me by way of a seeing them onassis map.",
        "time": "2014-08-25 00:37:13"
    },
    "comment2": {
        "content": "Buyer beware!.... I purchased this card last year and used the buy 1 get 1 free deal and was told it's meant for two people. This was at McConnell's fine ice cream on state street. This guy who's the manager or owner of the business said this deal is meant for you to bring someone along and enjoy the ice cream together and not for you to come in and walk away w/two ice cream cones and pay for one ice cream cone. At the end he said come back w/a friend. He was annoyed.",
        "time": "2020-10-09 16:54:26"
    }
}.
Check whether the two POIs are the same place. Notice that due to the errors, the latitude and longitude may be different although two POI represent the same place.
Please answer "Yes" or "No".
Answer: The answer is " |
| Answer | No |

Table 9: A data sample for URFR. The blue texts describe the question. The brown texts are the options. The teal texts denote the guidance that constrains the output of LLMs.

| | |
|---|---|
| Question | Question: Below is the coordinate information and related comments of a region, with the options of possible function for this region:
{
    "region": [(-90.0877900, 29.9689360), (-90.0872427, 29.9689360), (-90.0872427, 29.9696428), (-90.0877900, 29.9696428), (-90.0877900, 29.9689360)]",
    "pois": [
        {
            "latitude": 29.9694327,
            "longitude": -90.0874047,
            "comment1": {
                "content": "I cannot day enough about how much I love this place. NOCB popped up on my Instagram feed in 2017 with their Black Friday deals, signed up on a whim and never looked back. The classes are fun and exciting and a great way to get a feel for boxing and each of the trainers here. The gym has become my favorite past time and I love taking my friends in to understand why I'm hooked.",
                "time": "2019-12-05 01:43:01"
            },
            "comment2": {
                "content": "The best boxing gym in the city! I started boxing a year ago wanting to both get in better shape and learn the skills associated with boxing. I've tried a few places but ultimately settled at NOBC. The positive atmosphere is the first thing you notice about this gym, regardless of if you are a professional or a first time boxer everyone trains together and shares the same passion for boxing, wanting to better themselves through the sport. The gym has everything you need from a weight room, a full boxing ring/equipment, and a cardio/ab area. In a few short weeks training with the owner Chase I have become a better boxer. The gym is clean, friendly, and fun. I plan on training here for years to come.",
                "time": "2016-10-29 02:02:17"
            }
        }
    ],
    "options": "(0): Suburban Lake Area Neighborhood Park District (1): Suburban Pedestrian Oriented Corridor Business District (2): Historic Urban Neighborhood Business District (3): Greenway Open Space District (4): Historic Marigny Treme Bywater Commercial District"
}
Please answer which function the location is. Please just answer the number of your option with no other texts.
Answer: Option ( |
| Answer | 2): Historic Urban Neighborhood Business District |

### A.2.2 SPATIO-TEMPORAL REASONING

The dimension of spatio-temporal reasoning consists of four tasks: **Point-Trajectory Relationship Detection (PTRD)**, **Point-Region Relationship Detection (PRRD)**, **Trajectory-Region Relationship Detection (TRRD)** and **Trajectory Identification (TI)**. The data sample of point-trajectory relationship detection provides a trajectory and five points, then ask the model which point the trajectory passes through, as shown in Table 10. A sample of point-region relationship detection is given in Table 11, which ask the model to determine which region a point falls in according to the boundary lines of the regions and the coordinates of the point. As an enhancement to this task, trajectory-region relationship detection further ask which regions a trajectory passes through chronologically, as shown in Table 12. Trajectory identification aim to determine if two point sequences describe the same trajectory, whose data samples are constructed by four strategies, *i.e.*, downsampling, staggered sampling, spatial offset and temporal offset. By setting different downsampling rate or sampling

different points, we can get two point sequences that describe the same trajectory, as shown in Table 13. By adding spatial offset or temporal offset to the coordinates or timestamps of the trajectory, we can get another different trajectory, as shown in Table 14.

Table 10: A data sample for PTRD. The blue texts describe the question. The brown texts are the options. The teal texts denote the guidance that constrains the output of LLMs.

| | |
|---|---|
| Question | Question: The following is a sequence of points sampled from a trajectory and the meaning of each point is (longitude, latitude, timestamp): [(108.91226, 34.25924, 1477967031), (108.92136, 34.25929, 1477967109), (108.92268, 34.26271, 1477967184), (108.92247, 34.27329, 1477967274), (108.92732, 34.27659, 1477967352), (108.93702, 34.27663, 1477967430), (108.9435, 34.27682, 1477967505), (108.95271, 34.27686, 1477967586), (108.95937, 34.27675, 1477967662), (108.97203, 34.27726, 1477967767)]. The trajectory passes through one of the following points: (1) Point 1 (108.93244, 34.28307); (2) Point 2 (108.95336, 34.28628); (3) Point 3 (108.93661, 34.28624); (4) Point 4 (108.91681, 34.259265); (5) Point 5 (108.92387, 34.26896); Please answer which option the trajectory passes through. Answer: The trajectory passes through Point |
| Answer | 4 |

Table 11: A data sample for PRRD. The blue texts describe the question. The teal texts denote the guidance that constrains the output of LLMs.

| | |
|---|---|
| Question | Question: There are several regions, and the boundary lines of each region are presented in the form of a list of (longitude, latitude) below: Region 1: [(104.2483, 33.2447), (104.2481, 33.2440), (104.2470, 33.2438), (104.2466, 33.2440), (104.2464, 33.2443), (104.2463, 33.2446), (104.2477, 33.2456)] Region 2: [(104.2446, 33.2471), (104.2453, 33.2460), (104.2456, 33.2450), (104.2451, 33.2451), (104.2448, 33.2454), (104.2443, 33.2457), (104.2437, 33.2459), (104.2432, 33.2462), (104.2431, 33.2465)] Now there is a point with longitude 104.2444 and latitude 33.2460. Please directly answer the number of the region that this point falls in. Answer: The point falls in Region |
| Answer | 2 |

Table 12: A data sample for TRRD. The blue texts describe the question. The brown texts are the options. The teal texts denote the guidance that constrains the output of LLMs.

| | |
|---|---|
| Question | Question: There are several regions, and the boundary lines of each region are presented in the form of a list of (longitude, latitude) below:
Region 1: [(104.2483209, 33.2446592), (104.2480514, 33.2440436), (104.2469734, 33.2437741), (104.2465616, 33.2440436), (104.2464345, 33.2443130), (104.2462657, 33.2445689), (104.2477476, 33.2456337)]
Region 2: [(104.2446473, 33.2470611), (104.2452599, 33.2459617), (104.2456260, 33.2449552), (104.2450870, 33.2451215), (104.2448175, 33.2453910), (104.2442785, 33.2456605), (104.2437395, 33.2459300), (104.2432005, 33.2461995), (104.2430696, 33.2464690)]
Region 3: [(104.2476758, 33.2457578), (104.2459598, 33.2450870), (104.2454075, 33.2460224), (104.2447964, 33.2471191), (104.2465063, 33.2478088), (104.2476758, 33.2457578)]
Region 4: [(104.2445777, 33.2471861), (104.2427577, 33.2466877), (104.2423098, 33.2477300), (104.2424400, 33.2481779), (104.2433484, 33.2491652), (104.2433517, 33.2491689), (104.2436447, 33.2488824), (104.2442290, 33.2478118)]
Region 5: [(104.2464353, 33.2479333), (104.2447267, 33.2472441), (104.2443780, 33.2478698), (104.2438019, 33.2489228), (104.2436336, 33.2494729), (104.2451994, 33.2499855), (104.2458120, 33.2490264), (104.2464353, 33.2479333)]
Now there is a trajectory presented in the form of a list of (longitude, latitude): [(104.2453154, 33.2468798), (104.2431636, 33.2476642), (104.2448701, 33.2483024), (104.2427480, 33.2486476), (104.2466176, 33.2489308)]. Note that although we only provide the coordinates of some discrete points, the trajectory is actually continuous.
Please answer which regions it has passed through in chronological order: (1) [3, 2, 1, 5], (2) [3, 4], (3) [3, 4, 2, 3], (4) [3, 2, 4, 2, 1], (5) [3, 4, 5].
Answer only one option with no other texts. Answer: Option ( |
| Answer | 5): [3, 4, 5] |

Table 13: Data samples constructed by downsampling and staggered sampling for TI. The blue texts describe the question. The brown texts are the options. The teal texts denote the guidance that constrains the output of LLMs.

| Downsampling | |
|---|---|
| Question | Question: There are two point sequences and each sequence is sampled from a trajectory. The meaning of each point is (longitude, latitude, time stamp). Please answer whether these two sequences are sampled from the same trajectory. Sequence 1: [(108.91226, 34.25924, 1477967031), (108.92136, 34.25929, 1477967106), (108.92277, 34.26197, 1477967178), (108.92248, 34.27254, 1477967265), (108.92586, 34.27659, 1477967340), (108.93587, 34.27662, 1477967415), (108.94108, 34.27671, 1477967487), (108.95088, 34.27682, 1477967564), (108.95635, 34.27691, 1477967638)], Sequence 2: [(108.91226, 34.25924, 1477967031), (108.91715, 34.25925, 1477967067), (108.92136, 34.25929, 1477967106), (108.92275, 34.25931, 1477967142), (108.92277, 34.26197, 1477967178), (108.92257, 34.2661, 1477967217), (108.92248, 34.27254, 1477967265), (108.92307, 34.27581, 1477967301), (108.92586, 34.27659, 1477967340), (108.93109, 34.2766, 1477967379), (108.93587, 34.27662, 1477967415), (108.93958, 34.27668, 1477967451), (108.94108, 34.27671, 1477967487), (108.94591, 34.27739, 1477967523), (108.95088, 34.27682, 1477967564), (108.95329, 34.27687, 1477967602), (108.95635, 34.27691, 1477967638)]. You can confirm if their routes are the same by checking if sequence 1 passes through each point in sequence 2. Then, check if their timestamps are consistent. Finally, answer whether they are sampled from the same trajectory. Please answer "Yes" or "No". Answer: The answer is " |
| Answer | Yes |
| Staggered Sampling | |
| Question | Question: There are two point sequences and each sequence is sampled from a trajectory. The meaning of each point is (longitude, latitude, time stamp). Please answer whether these two sequences are sampled from the same trajectory. Sequence 1: [(108.91267, 34.25924, 1477967034), (108.91758, 34.25925, 1477967070), (108.92136, 34.25929, 1477967109), (108.923, 34.25931, 1477967145), (108.92273, 34.26228, 1477967181), (108.92256, 34.26648, 1477967220), (108.92248, 34.27273, 1477967268), (108.92317, 34.27594, 1477967304), (108.92621, 34.27659, 1477967343), (108.93137, 34.27661, 1477967382), (108.93614, 34.27663, 1477967418), (108.93984, 34.27668, 1477967454), (108.94133, 34.27671, 1477967490), (108.94635, 34.27738, 1477967526), (108.95162, 34.27684, 1477967568), (108.95344, 34.27687, 1477967605), (108.95665, 34.27689, 1477967641)], Sequence 2: [(108.91226, 34.25924, 1477967031), (108.91715, 34.25925, 1477967067), (108.92136, 34.25929, 1477967106), (108.92275, 34.25931, 1477967142), (108.92277, 34.26197, 1477967178), (108.92257, 34.2661, 1477967217), (108.92248, 34.27254, 1477967265), (108.92307, 34.27581, 1477967301), (108.92586, 34.27659, 1477967340), (108.93109, 34.2766, 1477967379), (108.93587, 34.27662, 1477967415), (108.93958, 34.27668, 1477967451), (108.94108, 34.27671, 1477967487), (108.94591, 34.27739, 1477967523), (108.95088, 34.27682, 1477967564), (108.95329, 34.27687, 1477967602), (108.95635, 34.27691, 1477967638)]. You can confirm if their routes are the same by checking if sequence 1 passes through each point in sequence 2. Then, check if their timestamps are consistent. Finally, answer whether they are sampled from the same trajectory. Please answer "Yes" or "No". Answer: The answer is " |
| Answer | Yes |

Table 14: Data samples constructed through spatial or temporal offset for TI. The blue texts describe the question. The brown texts are the options. The teal texts denote the guidance that constrains the output of LLMs.

| Spatial Offset | |
|---|---|
| Question | Question: There are two point sequences and each sequence is sampled from a trajectory. The meaning of each point is (longitude, latitude, time stamp). Please answer whether these two sequences are sampled from the same trajectory. Sequence 1: [(108.91226, 34.25924, 1477967031), (108.91715, 34.25925, 1477967067), (108.92136, 34.25929, 1477967106), (108.92275, 34.25931, 1477967142), (108.92277, 34.26197, 1477967178), (108.92257, 34.2661, 1477967217), (108.94056, 34.28908, 1477967265), (108.94115, 34.29235, 1477967301), (108.94394, 34.29313, 1477967340), (108.94917, 34.29314, 1477967379), (108.95395, 34.29316, 1477967415), (108.95766, 34.29322, 1477967451), (108.95916, 34.29325, 1477967487), (108.96399, 34.29393, 1477967523), (108.96896, 34.29336, 1477967564), (108.97137, 34.29341, 1477967602), (108.95635, 34.27691, 1477967638)], Sequence 2: [(108.91226, 34.25924, 1477967031), (108.91715, 34.25925, 1477967067), (108.92136, 34.25929, 1477967106), (108.92275, 34.25931, 1477967142), (108.92277, 34.26197, 1477967178), (108.92257, 34.2661, 1477967217), (108.92248, 34.27254, 1477967265), (108.92307, 34.27581, 1477967301), (108.92586, 34.27659, 1477967340), (108.93109, 34.2766, 1477967379), (108.93587, 34.27662, 1477967415), (108.93958, 34.27668, 1477967451), (108.94108, 34.27671, 1477967487), (108.94591, 34.27739, 1477967523), (108.95088, 34.27682, 1477967564), (108.95329, 34.27687, 1477967602), (108.95635, 34.27691, 1477967638)]. You can confirm if their routes are the same by checking if sequence 1 passes through each point in sequence 2. Then, check if their timestamps are consistent. Finally, answer whether they are sampled from the same trajectory. Please answer "Yes" or "No". Answer: The answer is " |
| Answer | No |
| Temporal Offset | |
| Question | Question: There are two point sequences and each sequence is sampled from a trajectory. The meaning of each point is (longitude, latitude, time stamp). Please answer whether these two sequences are sampled from the same trajectory. Sequence 1: [(108.91226, 34.25924, 1477967031), (108.91715, 34.25925, 1477967067), (108.92136, 34.25929, 1477967106), (108.92275, 34.25931, 1477967142), (108.92277, 34.26197, 1477967178), (108.92257, 34.2661, 1477967217), (108.92248, 34.27254, 1477967265), (108.92307, 34.27581, 1477967301), (108.92586, 34.27659, 1477967340), (108.93109, 34.2766, 1477967379)], Sequence 2: [(108.91226, 34.25924, 1478006153), (108.91715, 34.25925, 1478006189), (108.92136, 34.25929, 1478006228), (108.92275, 34.25931, 1478006264), (108.92277, 34.26197, 1478006300), (108.92257, 34.2661, 1478006339), (108.92248, 34.27254, 1478006387), (108.92307, 34.27581, 1478006423), (108.92586, 34.27659, 1478006462), (108.93109, 34.2766, 1478006501)]. You can confirm if their routes are the same by checking if sequence 1 passes through each point in sequence 2. Then, check if their timestamps are consistent. Finally, answer whether they are sampled from the same trajectory. Please answer "Yes" or "No". Answer: The answer is " |
| Answer | No |

Table 15: A data sample for DD. The blue texts describe the question. The brown texts are the options. The teal texts denote the guidance that constrains the output of LLMs.

| Question | Question: A has a longitude of 115.6249 and a latitude of 33.1811, while B has a longitude of 114.3897 and a latitude of 36.085839. Therefore, B is in the () from A. Please choose the correct answer from the following options and fill it in parentheses. (1) North, (2) Northeast, (3) East, (4) Southeast, (5) South, (6) Southwest, (7) West, (8) Northwest. Please directly give me the number of your option with no other texts. Answer: Option ( |
|----------|-----------------------------------------------------------------------------------------------------------------|
| Answer | 1) North |

Table 16: A data sample for NAV. The blue texts describe the question. The brown texts are the options. The teal texts denote the guidance that constrains the output of LLMs.

| Question | Question: There are 5 locations, numbered 0 to 4. There are some roads and each connects two locations. Each edge is a triplet consisting of the locations it connects and its length: Road 0: (location 0, location 4, 259.42 meters) Road 1: (location 0, location 1, 417.61 meters) Road 2: (location 1, location 4, 674.17 meters) Road 3: (location 1, location 3, 590.34 meters) Road 4: (location 1, location 2, 778.27 meters) Road 5: (location 2, location 3, 482.70 meters) Road 6: (location 2, location 4, 524.79 meters) All roads are bidirectional. Now, you are at location 1 and want to take the shortest path to location 2, which road should you choose? Options: (1) road 4, (2) road 3, (3) road 2, (4) road 1. Answer: The answer is ( |
|----------|-----------------------------------------------------------------------------------------------------------------|
| Answer | 1 |

### A.2.3 ACCURATE COMPUTATION

The assessing of accurate computation involves three tasks: **Direction Determination (DD)**, **Navigation (NAV)** and **Trajectory-Trajectory Relationship Analaysis (TTRA)**. Direction determination aim to predict the relative direction between two given coordinates, as shown in Table 15. Navigation gives a road network and ask LLMs to choose the edge that is on the shortest path from a source point to a destination point, as shown in Table 16. For trajectory-trajectory relationship analysis, two trajectories are given and the model is asked to count how many times they intersect, as shown in Table 17.

### A.2.4 DOWNSTREAM APPLICATIONS

We select three downstream applications for evaluation: **Flow Prediction (FP)**, **Trajectory Anomaly Detection (TAD)**, **Trajectory Classification (TC)** and **Trajectory Prediction (TP)**. Flow prediction asks LLMs to prediction the inflows/outflows according to the historical inflows and outflows, as illusatrated in Table **??**. As shown in Table 19 and Table 20, given a trajectory, trajectory anomaly detection and trajectory classification aims to infer if the trajectory is anomalous and the source of the trajectory, respectively. For trajectory prediction, the model is asked to predict the next point of a trajectory according to the historical points, as shown in Table 21.

Table 17: A data sample for TTRA. The blue texts describe the question. The brown texts are the options. The teal texts denote the guidance that constrains the output of LLMs.

| | |
|---|---|
| Question | Question: There are two trajectories presented in the form of a list of (longitude, latitude, timestamp) below:
trajectory A: [(104.24490, 33.24652, 1683618155), (104.24440, 33.24504, 1683619121), (104.24420, 33.24477, 1683620129), (104.24600, 33.24515, 1683621109), (104.24667, 33.24498, 1683622143)]
trajectory B: [(104.24458, 33.24707, 1683618164), (104.24242, 33.24675, 1683619137), (104.24375, 33.24676, 1683620199), (104.24522, 33.24833, 1683621179), (104.24615, 33.24663, 1683622182)]
Please calculate the number of times these two trajectories intersect, and choose your answer from following options:
(1) 2 times, (2) 3 times, (3) 4 times, (4) 0 times, (5) 1 times.
Note that two trajectories intersect if and only if they pass through the same point at the same timestamp. Give me your option with no other texts.
Answer: Option ( |
| Answer | 4) 0 times |

Table 18: A data sample for FP. The blue texts describe the question. The brown texts are the options. The teal texts denote the guidance that constrains the output of LLMs.

| | |
|---|---|
| Question | Question: Here is the historical data for taxi flow over 12 time steps in a specific region of Beijing:
Taxi inflows: [2.0, 1.0, 1.0, 2.0, 2.0, 0.0, 0.0, 1.0, 1.0, 0.0, 0.0, 0.0]
Taxi outflows: [2.0, 1.0, 1.0, 2.0, 2.0, 0.0, 0.0, 1.0, 1.0, 0.0, 0.0, 0.0].
The recording period for this data spans from 2015-11-01 22:30:00 to 2015-11-02 04:30:00, with each data point captured at 30-minute intervals. Note that the start time is inclusive, while the end time is exclusive.
Please forecast the taxi inflow for the subsequent 6 time steps during the period from 2015-11-02 04:30:00 to 2015-11-02 07:30:00, maintaining the same 30-minute interval for data recording. Please analyze the traffic patterns in this region, utilizing the given data or any additional information available, and generate the required predictions.
Answer: The predicted inflows for the 6 time steps are: [ |
| Answer | [0.0, 0.0, 1.0, 1.0, 0.0, 1.0] |

Table 19: A data sample for TAD. The blue texts describe the question. The brown texts are the options. The teal texts denote the guidance that constrains the output of LLMs.

| | |
|---|---|
| Question | Question: Below is a trajectory generated by a taxi, and each point in this trajectory is a tuple of (longitude, latitude, timestamp): [(108.91226, 34.25924, 1477967031), (108.91715, 34.25925, 1477967067), (108.92136, 34.25929, 1477967106), (108.92275, 34.25931, 1477967142), (108.92277, 34.26197, 1477967178), (108.92257, 34.2661, 1477967217), (108.92248, 34.27254, 1477967265), (108.92307, 34.27581, 1477967301), (108.92586, 34.27659, 1477967340), (108.93109, 34.2766, 1477967379), (108.93587, 34.27662, 1477967415), (108.93958, 34.27668, 1477967451), (108.94108, 34.27671, 1477967487), (108.94591, 34.27739, 1477967523), (108.95088, 34.27682, 1477967564), (108.95329, 34.27687, 1477967602), (108.95635, 34.27691, 1477967638), (108.96059, 34.27669, 1477967677), (108.96856, 34.277, 1477967737), (108.97323, 34.27733, 1477967776), (108.97674, 34.27742, 1477967812), (108.97917, 34.27868, 1477967854)]. The trajectory is anomalous if there is a detour, otherwise the trajectory is normal. Please answer if this trajectory is anomalous or normal. Please answer "This trajectory is normal" or "This trajectory is anomalous" with no other texts. Answer: This trajectory is |
| Answer | normal |

Table 20: A data sample for TC. The blue texts describe the question. The brown texts are the options. The teal texts denote the guidance that constrains the output of LLMs.

| | |
|---|---|
| Question | Question: The following is a sequence of points sampled from a trajectory, and the meaning of each point is (longitude, latitude, timestamp): [(116.3324016, 40.0743183, 1225573207), (116.3324566, 40.0743099, 1225573208), (116.3326216, 40.0742966, 1225573212), (116.3328333, 40.0742683, 1225573216), (116.3330533, 40.0742516, 1225573220), (116.3332683, 40.0742699, 1225573224), (116.3334999, 40.0742583, 1225573228), (116.3337183, 40.0742583, 1225573232), (116.3339750, 40.0742033, 1225573236), (116.3341916, 40.0742249, 1225573240), (116.3343866, 40.0742749, 1225573244), (116.3345883, 40.0743099, 1225573248), (116.3347933, 40.0742966, 1225573252), (116.3350016, 40.0743049, 1225573256), (116.3352266, 40.0743299, 1225573260), (116.3354566, 40.0743183, 1225573264), (116.3356466, 40.0743099, 1225573268), (116.3358816, 40.0743150, 1225573272)]. The trajectory is generated by one of the following option: (1) car, (2) bike, (3) pedestrian. Please calculate the length and the average speed of the trajectory, and answer which option is most likely to generate this trajectory. Answer: The trajectory is most likely to be generated by Option ( |
| Answer | 2 |

Table 21: A data sample for TP. The blue texts describe the question. The brown texts are the options. The teal texts denote the guidance that constrains the output of LLMs.

| | |
|---|---|
| Question | Question: Below is an ongoing trajectory generated by a taxi, and each point in this trajectory is a tuple of (longitude, latitude, timestamp): [(108.92788, 34.23136, 1477956224), (108.92637, 34.23206, 1477956254), (108.92599, 34.23226, 1477956284), (108.92527, 34.23263, 1477956314)]. Please predict the longitude and latitude of the next point. Answer: The longitude and latitude of the next point is |
| Answer | [108.92384, 34.23327] |

Table 22: The performance of *ACC* on sub-datasets of point-region relationship detection and trajectory-region relationship detection. $r$ denotes the number of regions and $l$ denotes the length of the trajectory.

| | PRRD | | | | TRRD | | | | |
|---|---|---|---|---|---|---|---|---|---|
| | $r = 2$ | $r = 3$ | $r = 4$ | $r = 5$ | $l = 2$ | $l = 4$ | $l = 6$ | $l = 8$ | $l = 10$ |
| ChatGPT | 0.9568 | 0.9176 | 0.8864 | 0.9352 | 0.0536 | 0.0312 | 0.0136 | 0.0160 | 0.0144 |
| GPT-4o | 0.9224 | 0.9160 | 0.9096 | 0.9272 | 0.2504 | 0.1088 | 0.0680 | 0.0624 | 0.0616 |
| ChatGLM2 | 0.5624 | 0.6144 | 0.4216 | 0.4880 | 0.3104 | 0.2736 | 0.2536 | 0.2880 | 0.2696 |
| ChatGLM3 | 0.9096 | 0.8400 | 0.7328 | 0.8152 | 0.2256 | 0.2144 | 0.2032 | 0.1784 | 0.1672 |
| Phi-2 | - | - | - | - | - | - | - | - | - |
| Llama-2-7B | 0.5888 | 0.6504 | 0.6208 | 0.7824 | 0.2128 | 0.2088 | 0.1936 | 0.2072 | 0.1944 |
| Vicuna-7B | 0.7840 | 0.7160 | 0.5920 | 0.7400 | 0.1864 | 0.2032 | 0.1832 | 0.2008 | 0.2104 |
| Gemma-2B | 0.7024 | 0.5696 | 0.5408 | 0.4848 | 0.2096 | 0.1904 | 0.2168 | 0.1960 | 0.1944 |
| Gemma-7B | 0.9056 | 0.9072 | 0.8904 | 0.9144 | 0.2096 | 0.1856 | 0.2128 | 0.1952 | 0.1928 |
| DeepSeek-7B | 0.8544 | 0.5968 | 0.5184 | 0.6000 | 0.1504 | 0.1328 | 0.1001 | 0.1088 | 0.0944 |
| Falcon-7B | 0.5602 | 0.4344 | 0.3296 | 0.3647 | 0.1995 | 0.2110 | 0.2090 | 0.2062 | 0.2046 |
| Mistral-7B | 0.5336 | 0.7104 | 0.7256 | 0.8696 | 0.1896 | 0.0704 | 0.0320 | 0.0304 | 0.0288 |
| Qwen-7B | 0.6448 | 0.5752 | 0.5184 | 0.5662 | 0.2544 | 0.1544 | 0.1312 | 0.1496 | 0.1408 |
| Yi-6B | 0.9192 | 0.8008 | 0.7560 | 0.8296 | 0.2184 | 0.1816 | 0.1744 | 0.1672 | 0.1816 |

# B EXPERIMENTAL DETAILS

## B.1 EVALUATED MODELS

We evaluate two closed-source models and a set of open-source models. The two closed-source models are **ChatGPT** (gpt-3.5-turbo-1106) and **GPT-4o** (gpt-4o-2024-05-13), both developed by OpenAI. For open-source models, we first select two models from the popular Llama family, *i.e.*, **Llama-2-7B** and **Vicuna-7B**, which are released by Meta and Large Model Systems Organization, respectively. Then, we include **Gemma-2B** and **Gemma-7B**, which are developed by Google DeepMind, based on Gemini research and technology. **Phi-2**, a model with only 2.7 billion parameters proposed by Microsoft Research, is evaluated to investigate the performance of lightweight language models. We also evaluate **ChatGLM2** and **ChatGLM3**, two open bilingual language models with 6B parameters based on General Language Model (GLM). Moreover, **Mistral-7B**, a large language model developed by Mistral AI, is also included. Futhermore, other baselines includes **Falcon-7B**, a LLM developed by Technology Innovation Institute; **Deepseek-7B**, the language model presented by Deepseek AI; **Qwen-7B**, the language model of Alibaba and **Yi-6B**, an open foundation model by 01.AI. All experiments about the open-source models are conducted on modelscope [7].

## B.2 DETAILED RESULTS

There are some tasks that consists of several sub-datasets. Specifically, for the point-region relationship detection task, we vary the number of regions from 2 to 5 to obtain 4 sub-datasets. In the trajectory-region relationship detection task, the trajectory length is set to 2, 4, 6, 8, 10 to construct five sub-datasets. Moreover, we adopt four strategies to construct the data samples for trajectory identification, resulting in four sub-datasets.

### B.2.1 BASIC PROMPT

The results on these sub-datasets with basic prompt are shown in Table 22, Table 23, Table 24 and Table 25. For point-region relationship detection, we observe that most models achieve higher performance on sub-datasets with fewer regions, which is in line with our intuition. But there are also exceptions, *e.g.*, Mistral-7B achieve higher performance with more regions. For the trajectory-region relationship detection, the performance of most models decreases with larger trajectory length, since longer trajectory makes the task more challenging. For trajectory identification, we observe that some

---

[7]https://github.com/modelscope/modelscope

Table 23: The performance of *ACC* on sub-datasets of trajectory identification.

| | Downsampling | Staggered | Temporal | Spatial |
|---|---|---|---|---|
| ChatGPT | 0.1784 | 0.0016 | 0.8464 | 0.3104 |
| GPT-4o | 0.1624 | 0.5840 | 0.0280 | 0.9920 |
| ChatGLM2 | 0.0000 | 0.0000 | 1.0000 | 1.0000 |
| ChatGLM3 | 0.9992 | 0.9368 | 0.8008 | 0.0000 |
| Phi-2 | 1.0000 | 1.0000 | 0.0000 | 0.0000 |
| Llama-2-7B | 0.1952 | 0.9992 | 1.0000 | 0.0000 |
| Vicuna-7B | 0.0000 | 0.0000 | 1.0000 | 1.0000 |
| Gemma-2B | 0.0000 | 0.0000 | 1.0000 | 1.0000 |
| Gemma-7B | 1.0000 | 1.0000 | 0.0000 | 0.0000 |
| DeepSeek-7B | 1.0000 | 0.9856 | 0.0000 | 0.0000 |
| Falcon-7B | 0.8264 | 0.0024 | 1.0000 | 1.0000 |
| Mistral-7B | 0.0056 | 0.0000 | 1.0000 | 0.7448 |
| Qwen-7B | 0.3992 | 0.3395 | 0.6047 | 0.5714 |
| Yi-6B | 0.9888 | 0.8856 | 0.0000 | 0.4144 |

Table 24: The performance of *ACC* on sub-datasets of navigation.

| | weighted edges | | | | unweighted edges | | | |
|---|---|---|---|---|---|---|---|---|
| node num | 5 | 6 | 7 | 8 | 8 | 9 | 10 | 11 |
| ChatGPT | 0.4704 | 0.4448 | 0.3952 | 0.3936 | 0.4784 | 0.4928 | 0.4352 | 0.3968 |
| GPT-4o | 0.8064 | 0.7024 | 0.6176 | 0.5712 | 0.8912 | 0.8880 | 0.8000 | 0.7648 |
| ChatGLM2-6B | 0.3136 | 0.2784 | 0.2736 | 0.2688 | 0.2960 | 0.3408 | 0.2768 | 0.2912 |
| ChatGLM3-6B | 0.2768 | 0.2464 | 0.2576 | 0.2528 | 0.2704 | 0.2496 | 0.2368 | 0.2704 |
| DeepSeek-7B | 0.3168 | 0.3152 | 0.2944 | 0.2720 | 0.3200 | 0.3056 | 0.3376 | 0.2848 |
| Falcon-7B | 0.2416 | 0.2640 | 0.2368 | 0.2256 | 0.2560 | 0.3024 | 0.2912 | 0.2704 |
| Gemma-2B | 0.2656 | 0.2688 | 0.2720 | 0.2384 | 0.2608 | 0.2384 | 0.2672 | 0.2624 |
| Gemma-7B | 0.4560 | 0.4432 | 0.3584 | 0.3200 | 0.4000 | 0.3936 | 0.3728 | 0.3648 |
| Llama-2-7B | 0.2848 | 0.3168 | 0.2752 | 0.3056 | 0.2528 | 0.2544 | 0.2528 | 0.2768 |
| Mistral-7B | 0.3600 | 0.2944 | 0.3248 | 0.3184 | 0.2848 | 0.2864 | 0.2704 | 0.2656 |
| Phi-2 | 0.3088 | 0.2816 | 0.2880 | 0.2656 | 0.3072 | 0.3088 | 0.2912 | 0.2784 |
| Qwen-7B | 0.3696 | 0.3456 | 0.3152 | 0.2960 | 0.2864 | 0.2928 | 0.2640 | 0.3152 |
| Vicuna-7B | 0.2576 | 0.2544 | 0.2656 | 0.2544 | 0.2512 | 0.2912 | 0.2544 | 0.2416 |
| Yi-6B | 0.3584 | 0.3680 | 0.3440 | 0.3120 | 0.3424 | 0.3456 | 0.3040 | 0.2944 |

models consistently answer "Yes" or "No", regardless of the question, *e.g.*, ChatGLM2 and Phi-2. We also observe that different models have different characteristics. For instance, GPT-4o can find out spatial offset in trajectories, but it failed to identify the temporal offset. ChatGLM3 is good at identifying downsampling, staggered sampling and temporal offset, but it did not recognize the spatial offset. No evaluated model can achieve high performance on all four sub-datasets. For navigation, the performance of most LLMs decreases with more nodes, since more nodes make the road network more complex. Given the same node number, most LLMs perform better on unweighted graphs than weighted graphs, the reason is that searching shortest path in weighted graphs costs more computing capacity to handle decimals in edge weights.

### B.2.2 IN-CONTEXT LEARNING

The results on sub-datasets of trajectory-region relationship detection with in-context learning are shown in Table 26. We find that in-context learning significantly improve the performance of ChatGPT on sub-datasets with the trajectory length of 2, but it is useless for sub-datasets with longer trajectories. We also observe that in-context learing slightly improve the performance of Gemma-2B on sub-datasets with trajectory length larger than 2, which is exactly opposite to ChatGPT.

Table 25: The performance of *MAE* and *RMSE* of flow prediction.

|  | inflow | | outflow | |
|---|---|---|---|---|
|  | MAE | RMSE | MAE | RMSE |
| ChatGPT | 39.50 | 44.46 | 35.16 | 39.96 |
| GPT-4o | 45.26 | 57.09 | 41.24 | 53.64 |
| ChatGLM2-6B | 65.96 | 70.28 | 61.48 | 65.50 |
| ChatGLM3-6B | 59.47 | 64.26 | 59.01 | 63.20 |
| DeepSeek-7B | 60.72 | 65.51 | 53.05 | 57.12 |
| Falcon-7B | 63.41 | 68.07 | 61.63 | 65.13 |
| Gemma-2B | 41.39 | 45.54 | 42.19 | 46.21 |
| Gemma-7B | 30.61 | 35.21 | 33.09 | 37.43 |
| Llama-2-7B | 61.21 | 65.35 | 46.37 | 50.09 |
| Mistral-7B | 40.05 | 44.43 | 35.47 | 42.59 |
| Phi-2 | 38.14 | 42.59 | 31.50 | 35.80 |
| Qwen-7B | 57.59 | 61.71 | 49.38 | 53.54 |
| Vicuna-7B | 49.16 | 53.25 | 47.22 | 51.03 |
| Yi-6B | 57.36 | 61.64 | 46.70 | 50.83 |

Table 26: The performance of *ACC* on sub-datasets of trajectory-region relationship detection with in-context learning, chain-of-thought prompting and fine-tuning. $l$ denotes the length of the trajectory.

|  | $l = 2$ | $l = 4$ | $l = 6$ | $l = 8$ | $l = 10$ |
|---|---|---|---|---|---|
| ChatGPT w/ ICL | 0.1432 | 0.0408 | 0.0120 | 0.0080 | 0.0088 |
| Llama-2-7B w/ ICL | 0.2000 | 0.1688 | 0.1328 | 0.1232 | 0.1376 |
| Gemma-2B w/ ICL | 0.2088 | 0.2472 | 0.2376 | 0.2384 | 0.2200 |
| ChatGPT w/ CoT | 0.7504 | 0.2520 | 0.1584 | 0.1112 | 0.0872 |
| Gemma-2B w/ CoT | 0.2210 | 0.2564 | 0.2287 | 0.1910 | 0.2125 |
| Gemma-2B w/ SFT | 0.7560 | 0.8104 | 0.8072 | 0.7512 | 0.7640 |

### B.2.3 CHAIN-OF-THOUGHT

The results on sub-datasets of trajectory-region relationship detection with chain-of-though prompting are shown in Table 26. We observe that CoT further significantly boost the performance of ChatGPT on most sub-datasets. With the trajectory length increases, the performance of ChatGPT with CoT decreases sharply. For Gemma-2B, CoT does not further improve its performance compared with ICL.

### B.2.4 FINE-TUNING

The results on sub-datasets of trajectory-region relationship detection after fine-tuning are shown in Table 26. We observe fine-tuning significantly improve the performace of Gemma-2B on all sub-datasets. The performance after fine-tuning does not decreases with larger trajectory length.