# OpenReview forum: "STBench: Assessing the Ability of Large Language Models in Spatio-Temporal Analysis"
_ICLR.cc/2025/Conference — ICLR 2025 Conference Withdrawn Submission_

### Official Review · Reviewer_KrKH · 2024-11-03

**Soundness:** 2
**Presentation:** 3
**Contribution:** 2
**Rating:** 5
**Confidence:** 4

**Summary:**

This paper proposes STBench to evaluate the spatiotemporal analysis ability of various LLMs. STBench includes 15 tasks that evaluate LLMs on four dimensions: knowledge comprehension, spatiotemporal reasoning, accurate computation, and downstream applications. A systematic evaluation has been performed over 14 LLMs on 15 tasks in three settings: in-context learning, chain-of-thought evaluation, and fine-tuning.

**Strengths:**

1. This paper provides a systematic evaluation of 15 LLMs on different spatiotemporal tasks. They are compared in three settings in-context learning, chain-of-thought evaluation, and fine-tuning.
2. The paper emphasizes the importance of spatiotemporal reasoning ability of LLMs which is usually ignored by many LLM studies.

**Weaknesses:**

1. The classification of tasks -- knowledge comprehension, spatiotemporal reasoning, accurate computation, and downstream applications -- is very vague. Some tasks can easily be classified into other categories. For example, why is trajectory prediction not a knowledge comprehension task? I cannot see the logic behind this classification.

2. After reading the proposed 15 tasks, many tasks are very wired and meaningless:

a) In the Urban Region Function Recognition task, why can we use the boundary line of regions and two randomly sampled POI to predict the urban function? First, the boundary line will not be helpful in this task. Second, two randomly sampled POIs can easily mislead the prediction. This is not a meaningful task.

b) In POI identification task, why do you add some disturbance to the POI? How does this magnitude of the disturbance affect the model performance?

3. For many tasks, such as Point-Trajectory Relationship Detection, Point-Region Relationship Detection, Trajectory-Region Relationship Detection, Direction Determination, Navigation, and Trajectory-Trajectory Relationship Analysis, we can use deterministic algorithms to solve this task. Why do we need to use LLMs? Can we just use LLM Agent to call some function to handle these tasks instead of letting LLMs to solve the tasks?

4. There are many citation errors in Section 2. "(Ji & Gao, 2023) evaluated the.. " should be "Ji & Gao (2023) evaluated the...".

5. I think this work is more suitable for the NeruIPS data & benchmark track instead of the ICLR research track.

**Questions:**

Please see the weakness.

---

### Official Review · Reviewer_caJj · 2024-11-04

**Soundness:** 2
**Presentation:** 3
**Contribution:** 2
**Rating:** 5
**Confidence:** 4

**Summary:**

The paper focuses on creating a benchmark to assess the LLM's ability in spatio-temporal tasks. The authors introduce STBench consisting of 15 tasks and ~70,000 QA pairs. Using this benchmark, the authors also evaluate 13 LLMs and provide some interesting insights regarding the performance LLMs on these tasks.

**Strengths:**

S1. The paper is well-written and easy to follow. Each task is described with details.

S2. The tasks in the created benchmark cover different aspects of an LLM, which are reasonable to me.

**Weaknesses:**

W1. The motivation of creating a new benchmark to evaluate LLMs' ability on spatio-temporal tasks is not convincing. The limitations of existing benchmarks (traditional ones and the ones designed for LLMs) are not discussed in details. What are unique tasks to LLMs?

W2. In related work, the authors stated that Roberts et al. introduce a benchmark specifically designed for multimodal models and is not applicable to single-modal LLMs. However, the authors also evaluate the multimodal models in the experiments. In fact, multimodal benchmarks better represent real-life scenarios.

W3. The data format described in Sec. 4.1 simplifies the tasks for LLMs. I understand that it makes evaluations more straightforward, but it is still a strong assumption, which is not likely to be true in real-life applications.

W4. The constructed benchmark essentially are created from a few different datasets. The downside of this choice is that there are no connections among these datasets, making each task independent. IMO, task dependencies bring additional challenges to LLMs, which can better assess an LLM's capability.

W5. The experimental evaluation are not solid. Specifically, ICL, CoT and fine-tuning evaluations do not have all models included. The chosen models are very different in their capabilities, which leads to the expected experimental results. My takeaway from the results is to always use the latest LLM for spatial-temporal tasks.

**Questions:**

Q1. How many data points are sampled from the Yelp dataset for POI category recognition?

Q2. The description of urban region function recognition is confusing. An example might be useful here.

Q3. In point-trajectory relationship detection, what is exactly the length of a shorter trajectory?

Q4. The prompt templates described in the paper seem to be generic. Is it fair to all LLMs?

Q5. Can you articulate the reasons to choose Gemma-2B and Llama-2-7B in the ICL evaluation? And why use Gemma-2B only in the CoT evaluation?

---

### Official Review · Reviewer_boU9 · 2024-11-05

**Soundness:** 2
**Presentation:** 2
**Contribution:** 2
**Rating:** 5
**Confidence:** 4

**Summary:**

The authors developed a new benchmark dataset to evaluate the spatial-temporal analysis capabilities of recent large language models (LLMs). They performed the analysis on four different categories Knowledge Comprehension, Spatio-Temporal Reasoning, Accurate Computation and Downstream Applications with in total 15 distinct tasks by using 13 LLMs (2 closed source and 11 open source models). One main insight is that LLMs perform poorly on many tasks probably due to the lack of spatio-temporal training examples during the pre-training phase but the performance increases when fine-tuning models or using prompt engineering techniques like few-shot prompting or Chain-of-Thought reasoning.

**Strengths:**

- The paper is well written with a clear structure and good to understand
- Creation of a spatial-temporal dataset is useful for further research

**Weaknesses:**

- Appendix for details is missing which would help understanding the set-up better
- There is a reference to missing training data but a more detailed interpretation of the results, based on the knowledge that LLMs have weaknesses in processing numerical operations and the assumption of missing training examples, would have been desirable for contextualizing the presented results.
- Difference between ChatGPT model version and GPT-4o is not clear

**Questions:**

Did fine-tuning increase the performance on all tasks? That might be an indicator that it is possible to train a "generalized" spatial-temporal LLM "expert".

**Details Of Ethics Concerns:**

-

---

### Official Review · Reviewer_QzJM · 2024-11-09

**Soundness:** 4
**Presentation:** 4
**Contribution:** 3
**Rating:** 8
**Confidence:** 3

**Summary:**

The paper proposes a new benchmark for testing the capabilities of LLMs on spatio-temporal analysis. The paper is well-written and easy to read. The benchmark defines four categories; knowledge comprehension, spatio-temporal reasoning, accurate computation and downstream analysis. They define multiple tasks for each category, total of 15 tasks. They provide a large collection of (70,000) QA pairs. The paper also conducts a detailed performance analysis using the benchmark on 13 LLMs, and show the strengths of the models on knowledge comprehension tasks. They also test fine-tuning, in-context-learning, and chain-of-thought reasoning and show that all these improve the performance. The benchmark is designed methodically and rigorously. It is extensive and will be a great resource to enhance the capabilities of LLMs for spatio-temporal analysis.

**Strengths:**

1. Methodically design benchmark covering different aspects of spati-temporal analysis
2. It includes extensive experiments, evaluating the capabilities of 13 different LLMs. The results show that model size is important for knowledge comprehension but not all tasks. Most evaluated models had difficulty in multi-step reasoning. Latest large models are better at math and computation and this reflects in spatio-temporal computation tasks as well.
3. The paper also shows how CoT, ICL and fine-tuning improves the model accuracy

**Weaknesses:**

1. Related work section needs better organization and clarification
2. It would have been also good to highlight the surprising results.
3. The tasks are all formulated as "Yes/No" or multiple choice questions, except the two prediction tasks. This looks very simple at first, but the experimental results show that even for such tasks the models struggle. It would be interesting to test different formulations of these problems and their impact on model accuracy.

**Questions:**

1. Please fix this sentence in Section 5.2

"We observe that ChatGPT and GPT-4o outperform poorer than most open source models on...."

2. The paper claims that "ChatGPT refuses to answer the questions of prediction". Can you clarify? Our experience is the models rarely says "I do not know"

---

### Note · Authors · 2024-11-25

I have read and agree with the venue's withdrawal policy on behalf of myself and my co-authors.